# Human neutralizing antibodies target a conserved lateral patch on H7N9 hemagglutinin head

Manxue Jia[1,7], Hanjun Zhao [2,3,7], Nicholas C. Morano[1,4,7], Hong Lu[1], Yin-Ming Lui[2], Haijuan Du[5], Jordan E. Becker[1,4], Kwok-Yung Yuen [2,3,6], David D. Ho [1], Peter D. Kwong [1,5], Lawrence Shapiro [1,4,5], Kelvin Kai-Wang To [2,3,6] ✉ & Xueling Wu [1] ✉

Avian influenza A virus H7N9 causes severe human infections with >30% fatality. Currently, there is no H7N9-specific prevention or treatment for humans. Here, from a 2013 H7N9 convalescent case in Hong Kong, we isolate four hemagglutinin (HA)-reactive monoclonal antibodies (mAbs), with three directed to the globular head domain (HA1) and one to the stalk domain (HA2). Two clonally related HA1-directed mAbs, H7.HK1 and H7.HK2, potently neutralize H7N9 and protect female mice from lethal H7N9/AH1 challenge. Cryo-EM structures reveal that H7.HK1 and H7.HK2 bind to a β14-centered surface and disrupt the 220-loop that makes hydrophobic contacts with sialic acid on an adjacent protomer, thereby blocking viral entry. Sequence analysis indicates the lateral patch targeted by H7.HK1 and H7.HK2 to be conserved among influenza subtypes. Both H7.HK1 and H7.HK2 retain HA1 binding and neutralization capacity to later H7N9 isolates from 2016–2017, consistent with structural data showing that the antigenic mutations during this timeframe occur at their epitope peripheries. The HA2-directed mAb H7.HK4 lacks neutralizing activity but when used in combination with H7.HK2 moderately augments female mouse protection. Overall, our data reveal antibodies to a conserved lateral HA1 supersite that confer neutralization, and when combined with a HA2-directed non-neutralizing mAb, augment protection.

H7N9 is an avian influenza A group 2 virus first transmitted to humans in the spring of 2013 most likely through live poultry market exposure in China[1–3]. The virus reemerged in the fall of 2013 and in the winter of later years, with the largest epidemic reported as the 5th wave in 2016–2017[4–6]. Though there is limited evidence for human-to-human transmission, a few mutations in the hemagglutinin (HA) gene of the virus might be sufficient to overcome its inefficiency for human transmission[7–10]. Like other influenza virus infections, the most common treatments against H7N9 are neuraminidase inhibitors Tamiflu (i.e., oseltamivir) and Relena (i.e., zanamivir), but oseltamivir-resistant

[1]Aaron Diamond AIDS Research Center, Columbia University Vagelos College of Physicians and Surgeons, New York, NY 10032, USA. [2]State Key Laboratory for Emerging Infectious Diseases, Carol Yu Centre for Infection, Department of Microbiology, School of Clinical Medicine, Li Ka Shing Faculty of Medicine, University of Hong Kong, Pokfulam, Hong Kong Special Administrative Region, China. [3]Centre for Virology, Vaccinology and Therapeutics, Hong Kong Science and Technology Park, Sha Tin, Hong Kong Special Administrative Region, China. [4]Department of Biochemistry, Zuckerman Mind Brain Behavior Institute, Columbia University, New York, NY 10027, USA. [5]Vaccine Research Center, National Institute of Allergy and Infectious Diseases, National Institutes of Health, Bethesda, MD 20892, USA. [6]Department of Clinical Microbiology and Infection, University of Hong Kong-Shenzhen Hospital, Shenzhen, Guangdong 518053, China. [7]These authors contributed equally: Manxue Jia, Hanjun Zhao, Nicholas C. Morano. ✉e-mail: kelvinto@hku.hk; xw2702@cumc.columbia.edu

strains have emerged[11–13]. Intravenous (i.v.) zanamivir, though not clinically approved, has been used on a compassionate basis in some severe cases because of favorable pharmacokinetics and in vitro susceptibility against oseltamivir-resistant strains[14,15]. Other antiviral treatment includes the endonuclease inhibitor Xofluza (i.e., baloxavir marboxil) that targets the viral polymerase and has been shown effective in mice against H7N9 lethal challenges[16]. Despite the use of neuraminidase inhibitors and endonuclease inhibitor, H7N9 case-fatality rate remains higher than 30%, and currently there is no licensed vaccine against H7N9 for humans. Concerns for a potential major outbreak warrant the development of human monoclonal antibodies (mAbs) against H7N9.

Because HA is the major target for influenza neutralizing antibodies, H7-reactive human mAbs have been isolated and characterized from H7N9 acute infections[17], convalescent cases[18], and H7N9 experimental vaccinees[19–21]. The binding sites of these mAbs have been mapped to the HA globular head (HA1) and stem (HA2) domains. A subset of HA1-directed mAbs potently neutralized H7N9 and protected mice from H7N9 challenges at doses of 0.3, 1, 5 mg/kg or higher[17–20]. These HA1-directed mAbs typically neutralized H7N9 by direct interference with or around the receptor (sialic acid) binding site[17,19,22]. These epitopes correspond to the antigenic sites A and B as previously mapped on the surface of H3 HA[23–25]. Of note, significant antigenic drift has been documented in the HA gene of 2016–2017 H7N9 from the initial 2013 isolates[17,26,27]. For example, Huang et al. isolated 17 neutralizing mAbs from four cases infected in 2013 and 2014, yet only three of these mAbs were active against viral isolates from 2016 to 2017[17]. A broad mAb FluA-20 targeting the HA1 trimer interface did not mediate neutralization in vitro, but protected mice from viral challenges by disrupting HA trimers and inhibiting cell-to-cell spread of virus[21]. HA2-directed mAbs have also demonstrated neutralizing activity against divergent subtypes[28–35], although typically not as robust in neutralizing activity when compared to HA1 mAbs. A few HA2 mAbs, neutralizing or not, protected mice from H7N9 challenges at 5 mg/kg[20], especially when engineered as mouse IgG2a, which has the highest Fc-mediated effector functions in mouse[36]. However, previous studies have not tested the combination of two or more mAbs that target different regions of H7N9 HA.

In the post COVID-19 era, preparedness for future pandemics has become a high priority, as exemplified by the science community closely monitoring a bird flu (avian influenza A H5N1) outbreak in US cows[37]. Here, we aim to facilitate the development of human mAbs against H7N9, which has also been considered one of the most serious pandemic threats. We obtained peripheral blood mononuclear cells (PBMCs) from a 2013 H7N9 convalescent case in Hong Kong with the virus isolated as A/Hong Kong/470129/2013 H7N9[14]. The course of this infection lasted for about one month and the treatment required extracorporeal membrane oxygenation (ECMO) and i.v. zanamivir[14]. Development of plasma neutralizing antibodies was evident at recovery. The PBMC sample used to isolate mAbs was collected one year post recovery. From the isolated mAbs, we not only demonstrate potent H7N9 neutralization, but use cryo-EM analysis to delineate a conserved lateral site-of-vulnerability in the HA head, a finding of vaccine relevance.

## Results
### H7-reactive mAb isolation
For H7-specific mAb isolation, we purchased a soluble recombinant H7 HA protein based on A/Shanghai/2/2013 H7N9 for biotinylation, followed by streptavidin-PE conjugation. With this H7-PE bait, we stained 5 million PBMCs from the H7N9_HK2013 donor and sorted a total of 68 IgG$^+$ B cells (defined as CD3$^-$CD19$^+$CD20$^+$IgG$^+$) that are H7-PE$^+$ (Fig. 1a). Most of the sorted cells were at the borderline of H7-PE staining, but a few stained brightly for H7-PE. From the sorted B cells, we performed

single B cell RT-PCR and recovered four H7-reactive mAbs—namely, H7.HK1, H7.HK2, H7.HK3, and H7.HK4.

Measured by ELISA, the four reconstituted mAbs bound tightly to the H7N9 HA antigen used for H7-PE staining (Fig. 1b). Pre-treating the H7N9 HA with Endoglycosidase H (Endo H) had no effect on the mAb binding profiles, indicating that these mAbs do not rely on H7 glycans for binding. After switching the ELISA coating antigen to HA1 of the matching H7N9 HA from A/Shanghai/2/2013, the binding curves of H7.HK1, H7.HK2, and H7.HK3 were fully retained, indicating that these mAbs bind to the globular head domain HA1; in contrast, H7.HK4 lost binding to H7N9 HA1, suggesting that its binding epitope is likely located in the HA2 stem domain (Fig. 1b). Because of the documented antigenic drift for 2016–2017 H7N9 isolates, we also tested the mAb binding to HA1s from A/Guangdong/17SF003/2016 and A/Hong Kong/125/2017 H7N9. The binding curves of H7.HK1, H7.HK2, and H7.HK3 to both 2016 and 2017 HA1s were fully retained, and H7.HK4 did not bind to any HA1s. In addition, all four mAbs bound tightly to a recombinant H7N7 HA antigen based on A/Netherlands/219/2003 H7N7 (Fig. 1b). To non-H7 HA proteins, H7.HK1 and H7.HK2 did not react with any of the tested non-H7 HA; H7.HK3 cross-reacted with H15N8 HA, and H7.HK4 cross-reacted with H10N8 and H15N8 HAs (Fig. 1b), which sequence-wise are the closest to H7 in group 2 influenza HA genes[38]. Western blot of a cleaved HA based on A/Shanghai/1/2013 H7N9 confirmed that H7.HK2 binds to HA1 and H7.HK4 binds to HA2 (Fig. 1c).

### H7-reactive mAb neutralization
Using expression plasmids separately encoding H7 and N9 genes from A/Shanghai/4664T/2013 to pseudotype with HIV-1 NL4-3Δ*env*.Luc backbone[39], we generated the H7N9 2013 pseudo virus and tested mAb neutralization by a luciferase readout from single round infection of Madin-Darby Canine Kidney (MDCK) cells (Fig. 1d). H7.HK1 and H7.HK2 each potently neutralized the H7N9 2013 pseudo virus with an IC$_{50}$ of 20 ng/mL, while the other two mAbs H7.HK3 and H7.HK4 did not neutralize at up to 10 µg/mL (Fig. 1d, Table 1). Similarly, we generated pseudo virus using the H7 from A/Hong Kong/125/2017 H7N9. Both H7.HK1 and H7.HK2 retained their neutralization titers against the H7N9 2017 pseudo virus with an IC$_{50}$ of 30 ng/mL, while the other two mAbs H7.HK3 and H7.HK4 did not neutralize (Fig. 1d, Table 1). We further assessed the mAb neutralization against three live replicating H7N9 viruses, A/Anhui/1/2013, A/Zhejiang/DTID-ZJU01/2013, and the donor's autologous isolate A/Hong Kong/470129/2013, for multiple rounds of infection in MDCK cells. Scored by the presence of cytopathic effect, mAbs H7.HK1 and H7.HK2 neutralized all three H7N9 live isolates with IC$_{50}$s ranging 0.26–1.0 µg/mL; however, they did not neutralize any non-H7N9 influenza isolates tested, indicating that these mAbs are specific to H7N9 (Table 1). The other two mAbs H7.HK3 and H7.HK4 did not neutralize any of the tested H7N9 and therefore were not tested against non-H7N9 viruses. The neutralization IC$_{50}$s of H7.HK1 and H7.HK2 using pseudo viruses were about 10-fold more potent than those using live replicating viruses, suggesting that the pseudo virus neutralization is more sensitive thus useful for initial screening of neutralizing mAbs, which could then be confirmed with live replicating viruses. Similar differences in IC$_{50}$ values have been reported for other HA-reactive mAbs tested by both pseudo virus and live replicating virus[29].

### Comparison of H7.HK2 to previous H7 HA1-directed neutralizing mAbs
Three previous mAbs representing the best from each corresponding study were compared to H7.HK2 for ELISA binding to H7 antigens and neutralization of H7N9 pseudo viruses. Cloned from plasmablasts of an acute infection in 2013–2014[17], mAb L4A-14 is directed at the receptor-binding site (RBS) and bound 2013 H7N9 HA1 and HA similarly to H7.HK2, retained full binding to 2016 and 2017 HA1s (Supplementary Fig. 1a) but lost vast majority of binding to H7N7 HA

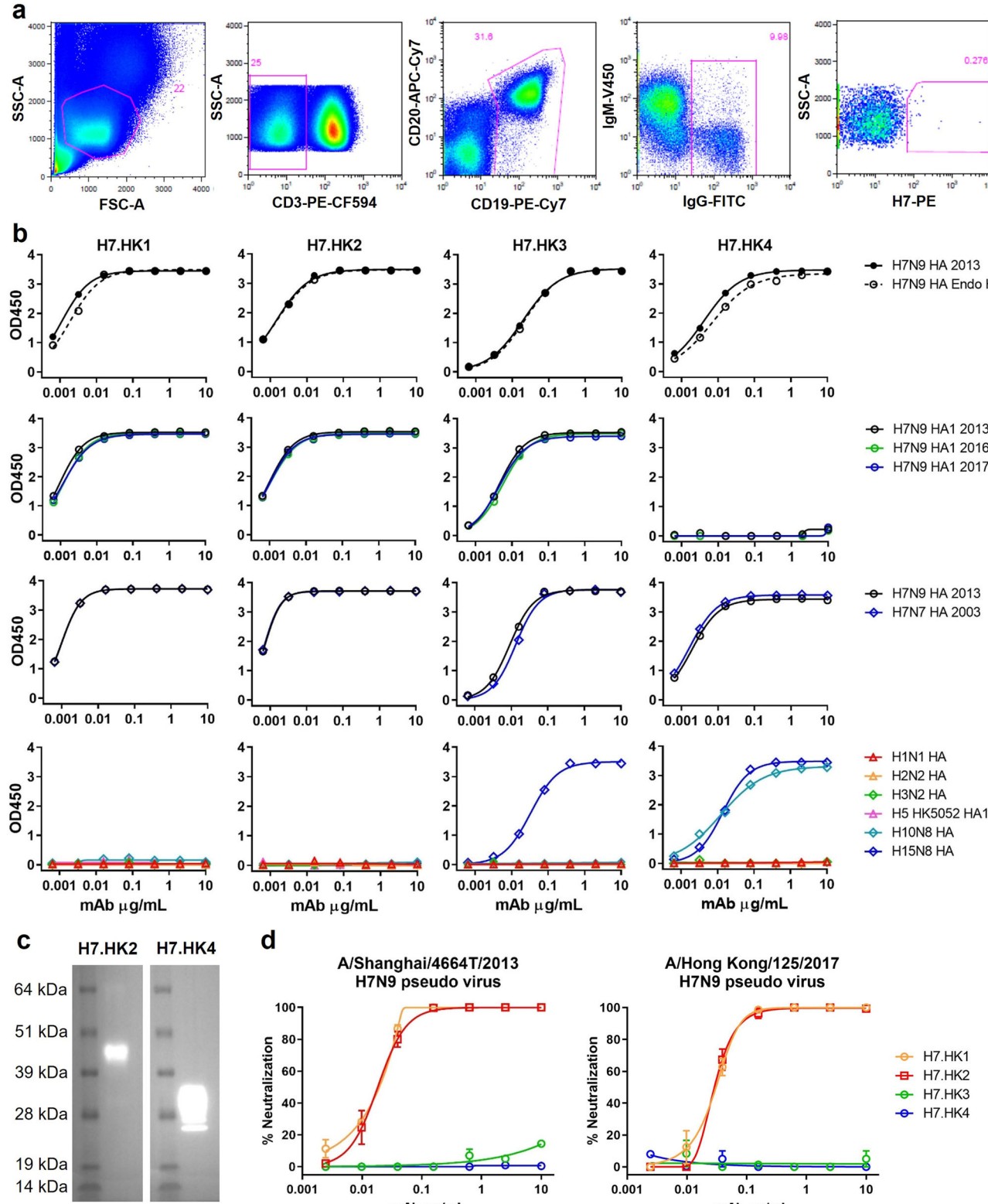

**Fig. 1 | Isolation and characterization of human H7N9 mAbs in vitro. a** FACS depicting the staining and selection of H7-specific B cells from donor H7N9_HK2013 PBMCs. SSC-A, side scatter area; FSC-A, forward scatter area. **b** ELISA binding curves of the indicated mAbs to soluble recombinant H7N9 HA, with and without Endo H treatment, to H7N9 HA1 from 2013, 2016, and 2017, to H7N7 HA, and to 6 non-H7 HA or HA1. **c** Western blot of a cleaved H7 HA (molecular mass of 43 kDa for HA1 and 30 kDa for HA2) with mAb H7.HK2 or H7.HK4. **d** Neutralization curves of H7.HK mAbs against H7N9 2013 and 2017 pseudo viruses in MDCK cells. Data shown are mean ± SEM. Source data are provided in the Source Data file. Similar results were independently reproduced at least once.

(Supplementary Fig. 1b). Derived from EBV transformed B cells after vaccination[19], another RBS-directed mAb H7.167 bound less well than H7.HK2 to 2013 H7N9 HA1 and HA and lost substantial binding to 2016 and 2017 HA1s and to H7N7 HA. Cloned from plasmablasts after vaccination[20], the non-RBS mAb 07-5F01 bound 2013, 2016, and 2017 H7N9 HA1s similarly to H7.HK2 and fully retained reactivity to H7N7 HA. Evaluated by pseudo virus neutralization (Supplementary Fig. 1c), the RBS-directed mAbs H7.167 and L4A-14 neutralized the 2013 H7N9 weaker than H7.HK2. L4A-14 neutralized the 2017 strain slightly better than 2013 H7N9, but H7.167's activity was further reduced by the 2017 virus. The non-RBS mAb 07-5F01 neutralized both 2013 and 2017 H7N9 viruses with comparable potency to H7.HK2. Hence, evaluated by H7 antigen binding and pseudo virus neutralization, H7.HK2 is superior to the two best previous RBS-directed mAbs L4A-14 and H7.167 and matches the one best previous non-RBS mAb 07-5F01 against H7N9.

## H7-reactive mAb sequences

Sequence analysis revealed that all four H7.HK mAbs are IgG1 (Table 2). H7.HK1 and H7.HK2 are clonal variants using IGHV4-59 for heavy chain with 8-10% somatic hypermutation (SHM) and a complementarity-determining region (CDR) H3 of 11 amino acids according to the Kabat and Chothia definition[40–42], and IGKV2-28 for light chain with 6% SHM and a CDR L3 of 9 amino acids. Though clonally related, H7.HK1 and H7.HK2 share only 3 out of 13-15 amino acid SHMs in the heavy chain V-gene and 1 out of 8 amino acid SHMs in the light chain V-gene (Supplementary Fig. 2). A putative N-linked glycosylation site is present in the light chain CDR L1 of H7.HK1 and H7.HK2. H7.HK3 uses IGHV7-4-1 for heavy chain with 7% SHM and a CDR H3 of 14 amino acids, and IGKV1-5 for light chain with 5% SHM and a CDR L3 of 8 amino acids. A putative N-linked glycosylation site is also present in H7.HK3 at the heavy chain CDR H2. H7.HK4 uses IGHV4-61 for heavy chain with 7%

SHM and a CDR H3 of 13 amino acids, and IGKV1-16 for light chain with 5% SHM and a CDR L3 of 9 amino acids (Table 2, Supplementary Fig. 2).

## H7-reactive mAb structures with HA trimer

For structural analysis, we expressed a soluble, disulfide-stabilized, and fully cleaved H7 HA trimer by transient transfection of Expi293F cells. H7.HK1 and H7.HK2 bound the H7 HA trimer tightly, H7.HK3 bound less well, and H7.HK4 did not bind at all (Supplementary Fig. 3a). As expected, the three previous neutralizing mAbs all bound the H7 HA trimer tightly, with H7.167 showing weaker binding (Supplementary Fig. 3b). We next generated the antibody fragments for antigen binding (Fabs) of H7.HK1 and H7.HK2 to bind the H7 HA trimer. We froze grids containing the Fab:HA complexes and determined cryo-EM structures of each Fab bound to an H7 HA trimer (Supplementary Fig. 4). A resolution of 3.62 Å for H7.HK1 and 3.69 Å for H7.HK2 was achieved (Fig. 2a, Supplementary Fig. 5, Table S1). These complex structures demonstrate that H7.HK1 and H7.HK2 are highly super-imposable (Fig. 2b) and their interactions with H7 are centered at β14 and extended to the surfaces of β10 and β19 (Fig. 2c). This β14-targeting surface partially overlaps with the antigenic site D towards sites A and B as previously mapped on H3[23,25]. Analysis of the H7.HK1 epitope demonstrates that most interactions are driven by the heavy chain and consist of seven hydrogen bonds (Y52:E121, R94:G124, G99:S167, D100:T126, Y100a:T165, Y100a:S167, S100c:T126) and one salt bridge (H53:E121) (Fig. 2d). The light chain is less involved in binding, making only one hydrogen bond (Y49:Q163) and weak hydrophobic interactions (Fig. 2e). The light chains of both H7.HK1 and H7.HK2 are glycosylated in CDR L1; this glycan plays no role in binding, but there is good density to support its presence. The epitope of H7.HK2 is similar to that of H7.HK1, only differing in slight contacts on the periphery (Supplementary Fig. 6a). In addition, nearly all hydrogen bonds are conserved between the two antibodies (Supplementary Fig. 6b). However, the substitution of F56S in CDR L2 of H7.HK2 results in an additional hydrogen bond with HA G129. This substitution also shifts the orientation of H7.HK2 CDR L2 slightly so that Y49 interacts with T165 for H7.HK2 instead of Q163 for H7.HK1 (Supplementary Fig. 6c). Finally, as H53 is substituted with tyrosine in the heavy chain of H7.HK2, it does not make the H53:E121 salt bridge.

To analyze the mechanism of neutralization, we first compared the binding site of H7.HK1 to that of four other H7-reactive antibodies with published structures, L4A-14, L4B-18, L3A-44 (PDB: 6II4, 6II8, 6II9)[17] and H7.167 (PDB: 5V2A)[19]. This analysis demonstrates that the binding site of H7.HK1 is almost completely distinct from that of these previously published antibodies, which compete for the RBS (Fig. 2f). The binding site of H7.HK1 is also distant from that of 07-5F01, which was mapped to an escape mutation R65K (corresponding to R57K here by H3 numbering) of HA1[20]. Direct competition ELISA applying biotin labeled H7.HK2 to bind H7 HA did not detect any effective competition by H7.HK3, H7.HK4, L4A-14, H7.167, and 07-5F01 (Fig. 2f), confirming the unique location of H7.HK1 and H7.HK2 epitopes. Further analysis of N = 1,483 H7 HA1 amino acid sequences from the Global Initiative on Sharing All Influenza Data (GISAID) revealed a conserved lateral patch (Fig. 2g) similar to what was initially identified in H1 viruses[43].

## Table 1 | Neutralization IC$_{50}$ of H7.HK mAbs against pseudo virus or live replicating virus

| Neutralization IC$_{50}$ (μg/mL) in MDCK cells | H7.HK1 | H7.HK2 | H7.HK3 | H7.HK4 |
|---|---|---|---|---|
| **Pseudo virus** | | | | |
| H7N9 A/Shanghai/4664T/2013 | 0.02 | 0.02 | >10 | >10 |
| H7N9 A/Hong Kong/125/2017 | 0.03 | 0.03 | >10 | >10 |
| **Live virus** | | | | |
| H7N9 A/Anhui/1/2013 | 0.26 | 0.26 | >30 | >30 |
| H7N9 A/Zhejiang/DTID-ZJU01/2013 | 0.26 | 1.0 | >30 | >30 |
| H7N9 A/Hong Kong/470129/2013 | 0.41 | 0.87 | ND | ND |
| H3N2 A/Hong Kong/400500/2015 | >30 | >30 | ND | ND |
| H1N1 A/Hong Kong/415742/2009 | >30 | >30 | ND | ND |
| H5N1 A/Hong Kong/459094/2010 | >30 | >30 | ND | ND |
| H5N1 A/Vietnam/1194/2004 | >30 | >30 | ND | ND |
| H9N2 A/Hong Kong/1073/1999 | >30 | >30 | ND | ND |

*ND* not done.

## Table 2 | Genetic composition, epitope, and neutralization function of H7.HK mAbs

| mAb ID | Origin | Time point | Isotype | V-gene (SHM%) | CDR3 length in amino acid | Epitope | Neutralization |
|---|---|---|---|---|---|---|---|
| H7.HK1 | Human | 1 year post recovery | IgG1 | HV4-59 (8%) KV2-28 (6%) | H3: 11, L3: 9 | H7 HA1 | Yes |
| H7.HK2 | Human | 1 year post recovery | IgG1 | HV4-59 (10%) KV2-28 (6%) | H3: 11, L3: 9 | H7 HA1 | Yes |
| H7.HK3 | Human | 1 year post recovery | IgG1 | HV7-4-1 (5%) KV1-5 (7%) | H3: 14, L3: 8 | H7 HA1 | No |
| H7.HK4 | Human | 1 year post recovery | IgG1 | HV4-61 (7%) KV1-16 (5%) | H3: 13, L3: 9 | H7 HA2 | No |

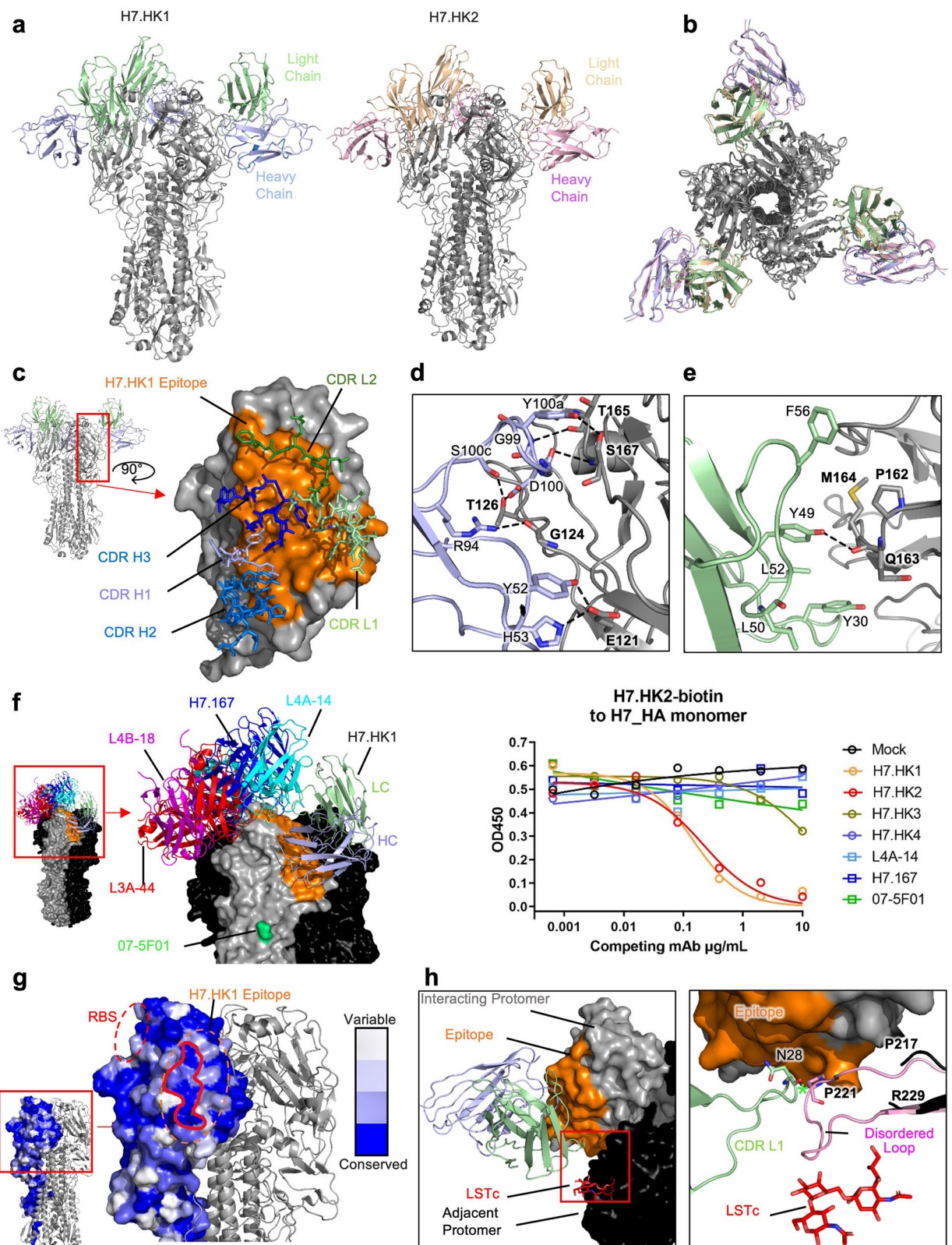

Strikingly, the lateral patch epitope of H7.HK1 was distal to the RBS of the protomer with which it interacted and closer to the RBS on the adjacent protomer. To further examine the relationship between the mAb binding site and RBS, the human receptor analogue Sialylneolacto-N-tetraose c (LSTc) was modeled into the RBS of H7 (PDB: 4BSE)[44] in the H7.HK1 complex. Interestingly, no steric clashes were observed between H7.HK1 and sialic acid bound to the adjacent

protomer, and no mAb interaction with RBS (Fig. 2g). However, the HA 220-loop (G218-G228), which makes hydrophobic contacts with sialic acid has no density present in the structure of H7.HK1 or H7.HK2 bound to HA, suggesting that these antibody binding causes the 220-loop to become disordered. All previously mentioned H7 structures (Fig. 2f), as well as an additional cryo-EM structure in which Fab 1D12 is bound to the stem region of H7 HA (PDB: 6WXL)[35] have consistent

**Fig. 2 | Structural analysis of H7.HK1 and H7.HK2 in complex with H7 HA trimer.** **a** Cryo-EM structures of H7.HK1 and H7.HK2 bound to H7 HA in the head region. **b** Top view of alignment of H7.HK1 and H7.HK2 complex structures. **c** Surface presentation of the H7.HK1 epitope (orange) on H7 HA1, with interacting CDRs shown. **d** H7.HK1 heavy chain forms seven hydrogen bonds and one salt bridge with H7 HA1. **e** H7.HK1 light chain forms one additional hydrogen bond with H7 HA1, and the interactions are stabilized by hydrophobic residues on the periphery of the light chain interface. **f** Modeling published structures of H7 HA1-binding antibodies (PDB: 6II4, 6II8, 6II9, 5V2A) onto the H7.HK1 bound structure, with an escape mutation R57K (green) reported for mAb 07-5F01. Competition ELISA with biotinylated H7.HK2 binding to the H7 HA monomer, in which unlabeled competing mAbs were titrated at increasing concentrations to evaluate the effect on H7.HK2 binding. Source data is provided in the Source Data file. **g** Sequence analysis of N = 1,483 H7 HA1s revealed a conserved lateral patch that largely overlaps with the H7.HK1 epitope. **h** Modeling the binding site of human receptor analogue LSTc (red) based on a previous crystal structure (PDB: 4BSE) onto H7 from the H7.HK1 complex, showing that H7.HK1 does not compete with sialic acid on the adjacent protomer (black). Alignment of the H7.HK1 complex with a previous crystal structure of H7 (PDB: 4BSE) shows that the 220-loop (pink) required for sialic acid binding (G218-G228) is disordered and would clash with the H7.HK1 light chain if it were present. Green asterisk symbol denotes the <2 Å clash between the CDR L1 N28 and the predicted location of P221 on HA1.

electron density accounting for this loop. Alignments of the H7.HK1 complex structure with the crystal structure of H7 HA bound to LSTc (PDB: 4BSE)[44] demonstrate where the 220-loop would be when receptor is bound and that the light chain of H7.HK1 would clash with this loop (Fig. 2h), further supporting that H7.HK1 and H7.HK2 act by causing the 220-loop to become disordered, thus preventing its interactions with the sialic acid receptor. The HA1 trimer interface mAb FluA-20 interacts with the non-RBS side of 220-loop on the protomer it interacts with[21]. To our knowledge, this mechanism of neutralization employed by H7.HK1 and H7.HK2—disrupting the 220-loop on H7 trimer—is distinct from previously reported HA1-directed H7N9 neutralizing mAbs, which all directly compete with sialic acid for binding to HA on the protomer they interact with[17,19,21,22,28].

Since the H7N9 HA gene has significantly evolved and changed in 2016–2017 compared to that of 2013 (with up to 12 amino acid substitutions in HA1), we examined the locations of mutated residues in the epitopes of H7.HK1 and H7.HK2 that consist of 32 contacting residues in HA1 for both mAbs (Supplementary Fig. 7a). There are three mutations in the binding site of H7.HK1 and H7.HK2—namely, A122T/P, S128N, and R172K, appeared in 2016–2017 compared to the 2013 H7N9, and all three mutations are located at one side edge of the epitopes (Supplementary Fig. 7b), thus not altering the mAb interactions with HA1. This analysis is consistent with the intact binding of H7.HK1 and H7.HK2 to both 2016 and 2017 HA1s aligned to the 2013 HA1 (Fig. 1b) and the mAbs' retention of neutralization against the H7N9 2017 pseudo virus (Fig. 1d).

### H7-reactive mAb mouse protection
We next assessed the prophylactic and therapeutic effect of H7.HK mAbs as human IgG1 in a mouse lethal challenge model. To assess mAb prophylactic effect, BALB/c mice (n = 5–10 per group from 1 to 2 experiments) were injected intraperitoneally (i.p.) with human H7N9 mAbs one day before intranasal (i.n.) challenge of 10-fold 50% lethal dose (10 LD$_{50}$) of A/Anhui/1/2013 H7N9 virus. Given 100 µg per mouse (equivalent to 5 mg/kg), the neutralizing mAbs H7.HK1 and H7.HK2 each fully protected mice without apparent weight loss (Fig. 3a); given 20 µg per mouse (equivalent to 1 mg/kg), H7.HK2 still fully protected mice from death (defined as ≥20% weight loss), with up to 8% average weight loss; H7.HK1 protected 7 out of 10 mice from death, with up to 12% average weight loss for mice that survived (Fig. 3a). By day 2 post challenge, the weight preservation was significantly better in mice receiving 20 µg of H7.HK1 or H7.HK2 than mice receiving the placebo mAb or phosphate buffered saline (PBS). Mice receiving the non-neutralizing mAbs H7.HK3 or H7.HK4 (100 µg or 20 µg) were not protected and showed no difference from placebo mAb and PBS controls (Fig. 3a).

Since anti-HA2 mAbs have demonstrated Fc-mediated protection against influenza[45], we converted the anti-HA2 non-neutralizing mAb H7.HK4 to mouse IgG2a—an isotype that mediates strong Fc effector function in mice, and tested it for prophylaxis in the mouse challenge model, along with mouse IgG1, which lacks Fc effector function in mice[36]. Given 100 µg per mouse, H7.HK4 mouse IgG2a but not IgG1 protected 4 out of 5 mice from death, with up to 17% average weight loss for mice that survived (Fig. 3a). By day 3 post challenge, the weight

preservation was significantly better in mice receiving H7.HK4 mouse IgG2a than mice receiving H7.HK4 mouse IgG1 or placebo mouse IgG2a. Though survived, mice receiving 100 µg H7.HK4 mouse IgG2a lost more weight than those receiving 20 µg neutralizing mAbs H7.HK1 or H7.HK2 (Fig. 3a), indicating less prophylaxis efficiency for H7.HK4 than H7.HK1 and H7.HK2.

Since the H7.HK2 and H7.HK4 mAbs bind to different sites on the HA and protect through different mechanisms, we tested the combination of suboptimal dose of 20 µg H7.HK2 (as human IgG1) with 100 µg H7.HK4 mouse IgG2a in the mouse challenge model, using 20 µg H7.HK2 (as human IgG1) with 100 µg H7.HK4 mouse IgG1 as a control. Compared to this control group, which protected 9 out of 10 mice from death and lost up to 11% body weight for mice that survived, the combination of 20 µg of H7.HK2 (as human IgG1) with 100 µg H7.HK4 mouse IgG2a fully protected mice from death, with only up to 7% weight loss, and the weight difference was statistically significant between these two groups since day 3 post challenge (Fig. 3a), indicating a beneficial role of H7.HK4 in mAb combination regimen. Overlaying the survival and body weight data of the 20 µg H7.HK2 alone group from the previous experiment (Fig. 3a), H7.HK2 in combination with H7.HK4 mouse IgG2a did not improve the body weight trough from day 4–6 post challenge as both groups fully protected mice from death with up to 7-8% weight loss; the mAb combination demonstrated a statistical trend and then significance for improved recovery of weight loss starting on day 7 post challenge (Fig. 3a).

To assess mAb therapeutic effects, we first i.n. challenged mice (n = 5–10 per group from 1 to 2 experiments) with 10 LD$_{50}$ of A/Anhui/1/2013 H7N9 virus, waited for one day, and then on day 1 post challenge i.p. injected mice with 100 µg H7.HK1 or H7.HK2 as human IgG1, or H7.HK4 as mouse IgG2a (Fig. 3b). Twelve and 13 out of 15 mice receiving 100 µg H7.HK1 or H7.HK2 one day after viral challenge initially lost weight similarly to placebo and PBS controls but then started to recover on day 5 after challenge. Therefore, the neutralizing mAbs H7.HK1 and H7.HK2 showed both prophylactic and therapeutic efficacies in the mouse lethal challenge model. None of the 5 mice receiving 100 µg H7.HK4 mouse IgG2a one day after challenge survived (Fig. 3b), indicating that H7.HK4 as mouse IgG2a demonstrated measurable prophylactic effect but not therapeutic efficacy.

### Comparison of H7.HK1/2 to previously published lateral patch antibodies
Thus far, there have been two other published structures of lateral patch-binding antibodies on the HA head, 045-09-2B05 (PDB: 7MEM) and Fab6649 (PDB: 5W6G)—both of which bind H1[43,46]. Comparing 045-09-2B05, Fab6649, and H7.HK1 demonstrated diverse angles of approach and different heavy and light chain orientations towards the lateral patch (Fig. 4a). The epitopes of these three antibodies were all centered on the lateral patch, with the composite footprint of all three defining a lateral patch supersite of vulnerability (Fig. 4b). The epitope of H7.HK1 was most similar to that of Fab6649. However, the light chain of H7.HK1 was in a slightly higher position on the head of HA than the heavy chain of Fab6649 (Fig. 4a, b), which allowed CDR L1 to clash with the 220-loop. H7.HK1 heavy chain overlapped with the light chain

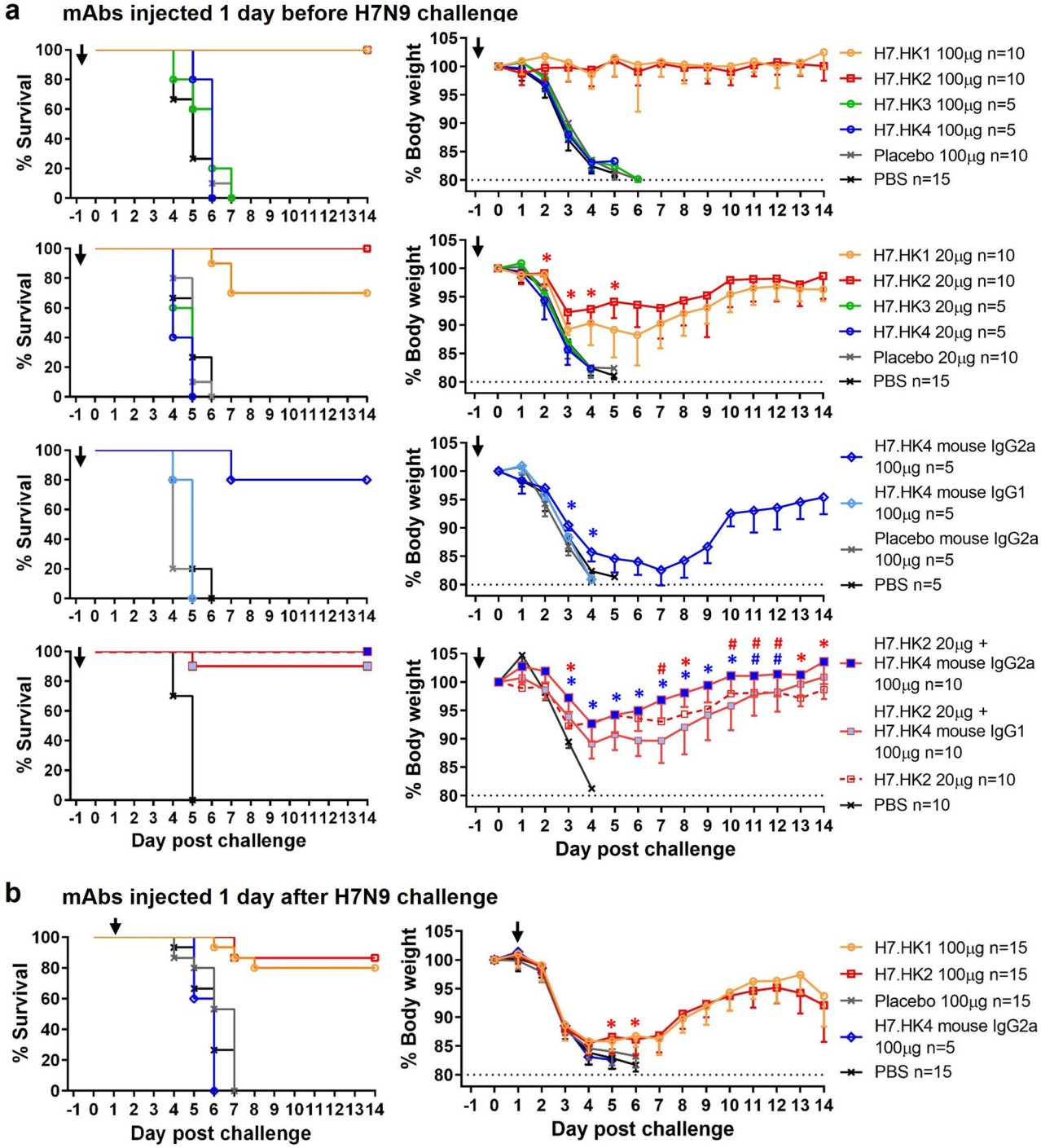

**Fig. 3 | Prophylactic and therapeutic effects of human H7N9 mAbs in mice i.n. challenged with 10 LD$_{50}$ of A/Anhui/1/2013 H7N9. a** Female mice were i.p. injected 100 μg (equivalent of 5 mg/kg) or 20 μg (equivalent of 1 mg/kg) of the indicated mAbs (as human IgG1 unless otherwise specified) one day before viral challenge; % survival (<20% weight loss) and % body weight of survived mice were plotted over time. **b** Female mice were i.p. injected 100 μg of the indicated mAbs one day after viral challenge; % survival and % body weight of survived mice were plotted over time. Arrows indicate the time when mAbs were administered. Control groups of a non-H7 placebo mAb and PBS were included. Data for each group were combined from 1 to 2 experiments and shown as mean – SEM. Source data with $P$ values from two-sided unpaired student's t-test are provided in the Source Data file. Asterisk symbols denote $P < 0.05$, and # denote $P < 0.1$.

of 045-09-2B05, and the heavy chain of 045-09-2B05 occupied an epitope distinct from H7.HK1, H7.HK2, and Fab6649. The epitopes of 045-09-2B05 and Fab6649 had modest overlap, centered around the conserved lateral patch (Fig. 4c colored in magenta). Of this overlapping epitope surface, there were four residues conserved between H1 and H7 (positions E121, S/T126, Y168, and R/K172). The overall structure of this site of vulnerability was also conserved between H1

and H7 (Fig. 4c). Comparison of the interactions between Fab6649, 045-09-2B05, H7.HK1, H7.HK2, and these four conserved HA residues revealed different modes of recognition for each antibody (Fig. 4d). Thus, the lateral patch supersite, as defined by Fab6649, 045-09-2B05, H7.HK1, and H7.HK2, was composed in part by residues that were conserved in H1 and H7 and could be targeted via diverse chemistries and modes of recognition.

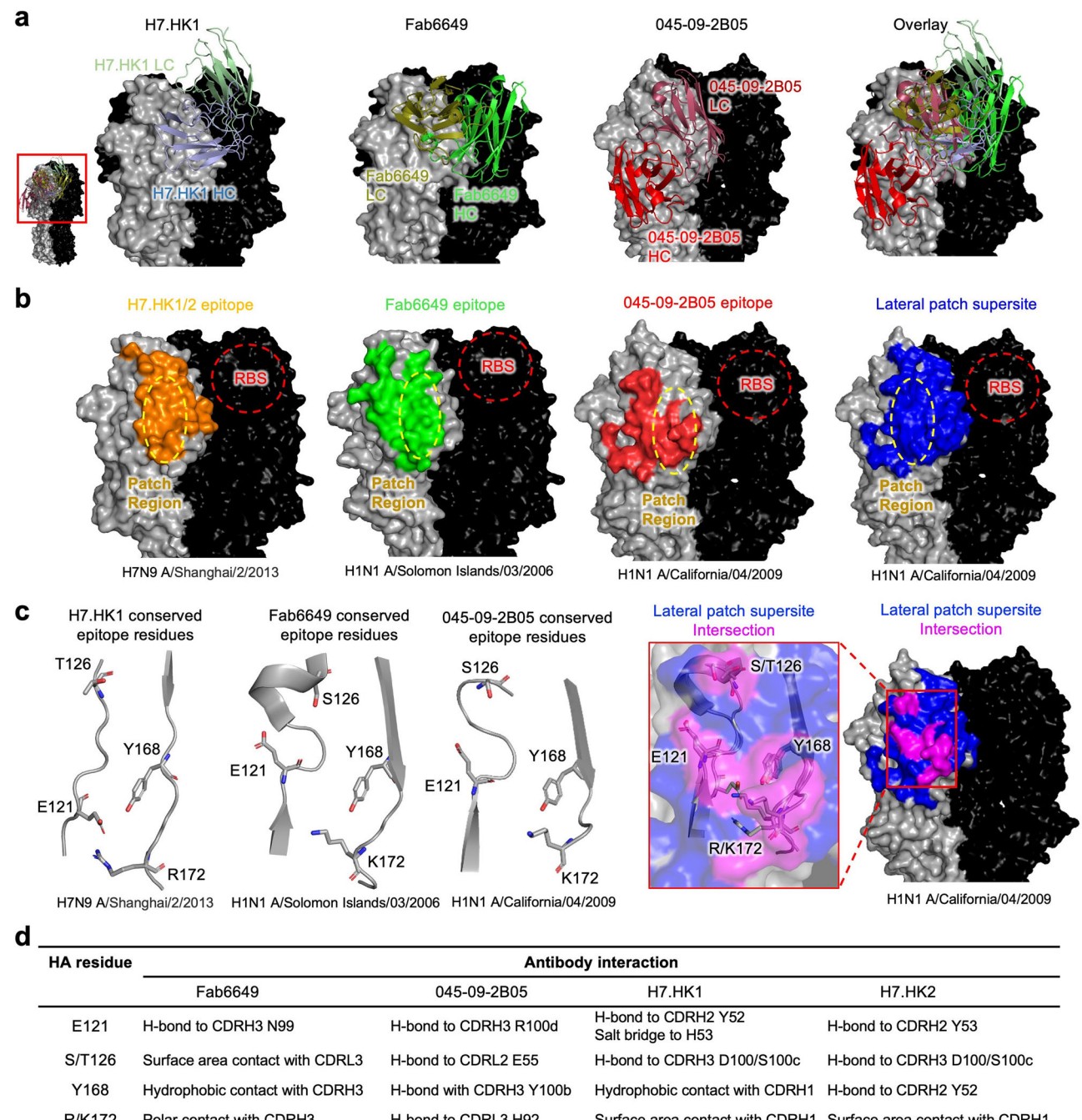

**Fig. 4 | H7.HK1 and two other lateral patch-binding antibodies define a conserved supersite of vulnerability on the HA head. a** Representation of H7.HK1, Fab6649, and 045-09-2B05 bound to their respective HAs indicates diverse angles of approach and heavy/light chain orientations towards the lateral patch supersite. **b** Comparison of epitopes of H7.HK1, Fab6649, and 045-09-2B05 centering on the lateral patch defines the lateral patch supersite (blue). **c** A subset of epitope surface, centered on the lateral patch, overlaps between H7.HK1, Fab6649, and 045-09-2B05 (magenta). Of shared epitope surface, a subset of epitope residues is conserved (positions 121, 126, 168, and 172 by H3 numbering). **d** The four lateral patch antibodies contact conserved residues using diverse chemistry. Table displays the type of interaction between each residue and antibody.

## Discussion

Already endemic, adapted, and evolved in humans for 11 years, H7N9 continues to pose risk and infect humans exposed to infected poultry in China. While the current risk to public health is low, the pandemic potential of H7N9 is especially concerning if it were to gain the ability of sustained human-to-human transmission. Based on its biological features such as dual affinity for avian and human receptors, high case-fatality rate, resistance to neuraminidase inhibitors, and lack of pre-existing immunity in the human populations, there is an immediate need and interest to develop human mAb prophylaxis and therapeutics against H7N9, to which a specific treatment or licensed vaccine (for humans) is not available.

In this study, we identified two HA1-directed clonally related human mAbs H7.HK1 and H7.HK2 that neutralized H7N9 with potencies and mouse protection efficacies (prophylactic and therapeutic) in line with the best of previous H7N9 mAbs. Specifically, a combined phage library from three H7N9 convalescent cases yielded a single neutralizing mAb clone[18], represented by the best clonal member,

HNIgGA6, which neutralized H7N9 and protected mice against a lethal challenge at 5 mg/kg with up to about 10% weight loss[18]. Likewise, from a study of four H7N9 acutely infected cases, the best mAb L4A-14 cloned from plasmablasts protected mice against a lethal challenge at 10 mg/kg with up to about 10% weight loss[17]. The most potent mAb H7.167 from a study of EBV transformed B cells from five representative H7N9 experimental vaccinees neutralized H7N9 and protected mice against a sub-lethal challenge of H7-PR8 at 1.65 mg/kg without apparent weight loss[19]. The best HA1-directed neutralizing mAb 07-5F01 from a study of H7N9 experimental vaccinees' plasmablasts protected mice against a lethal challenge at 0.3 mg/kg without apparent weight loss[20]. The broad HA1 trimer interface mAb FluA-20 from a healthy donor with extensive influenza vaccinations lacked in vitro neutralization but protected mice against a sub-lethal challenge of H7-PR8 at 10 mg/kg without apparent weight loss[21]. In comparison, H7.HK1 and H7.HK2 protected mice against a lethal challenge at 1 mg/kg with up to 12% weight loss.

We have also structurally defined the epitopes of H7.HK1 and H7.HK2 to the β14-centered surface of H7 HA1, partially overlapping with antigenic site D, targeting a lateral patch[43] rather than the commonly targeted RBS and trimer interface by previous H7N9 mAbs[28]. Structural comparisons demonstrated that H7.HK1 and H7.HK2 interacted with H7 differently from L4A-14, H7.167, 07-5F01, and FluA-20. By escape mutations, a previous H3 neutralizing mAb D1-8 was mapped to the lower part of antigenic site D towards site E[47]; this epitope partially overlaps with the H7.HK1 and H7.HK2 epitope described here. However, without structural data, the action of neutralization by D1-8 cannot be determined. Importantly, D1-8 does not react to H7, and likewise, H7.HK1 and H7.HK2 do not react to H3. Hence, D1-8 cannot replace the anti-H7N9 function of H7.HK1 and H7.HK2. The unique β14-targeting epitope on the HA1 lateral patch would render H7.HK1 and H7.HK2 favorable candidates for combination prophylaxis and therapy against H7N9 to augment protection efficacy and increase the genetic barrier for viral escape. This is supported by data showing no competition between H7.HK1 and H7.HK2 with two RBS and one non-RBS directed antibodies (Fig. 2f). Indeed, the H1 lateral patch has been a promising target for next-generation H1 vaccine strategy as previous studies showed that H1 lateral patch-binding antibodies are abundant in humans, and that they react broadly with H1 viruses and cross react with some H3 viruses[46,48]. Our work here demonstrates that the lateral patch is a viable epitope for H7 vaccine and therapeutic antibody development as well. Previously reported lateral patch antibodies are all restricted to expressing IGHV3 or IGHV4-39 genes and often had a Y-x-R motif in CDR H3[43,46]. H7.HK1 and H7.HK2 are derived from IGHV4-59 and do not contain a Y-x-R motif in CDR H3. In addition, unlike other reported lateral patch antibodies, H7.HK1 and H7.HK2 disrupt the structure of the 220-loop of the H7 RBS. Therefore, the lateral patch binding site of HA1 is expanded to H7 and composed in part with residues conserved between H1 and H7, which makes it a supersite of vulnerability that could be targeted by more diverse antibodies than previously recognized.

H7N9 has evolved over time and its HA gene has significantly changed in 2016–2017 compared to that of 2013. Consequently, most neutralizing mAbs isolated from individuals infected or vaccinated with the 2013 H7 HA lost reactivity to 2016–2017 isolates, requiring updated H7 immunogens for mAb and vaccine development[17]. We show that three mutations appeared in 2016–2017 are located at the periphery of the H7.HK1 and H7.HK2 epitopes and confirmed that the binding profiles of H7.HK1 and H7.HK2 are intact to both 2016 and 2017 HA1s as compared to 2013 HA1. We also showed that H7.HK1 and H7.HK2 retained their neutralization titers against the H7N9 2017 pseudo virus. The previous RBS-directed mAb HNIgGA6 was shown to lose reactivity to V186G and L226Q mutations[18] that are present in A/Netherlands/219/2003 H7N7 and A/Guangdong/17SF003/2016 H7N9, respectively. Evaluated by both antigen binding and pseudo

virus neutralization, H7.HK2 is superior to the two best previous RBS-directed mAbs L4A-14 and H7.167 and matches the one best previous non-RBS mAb 07-5F01 against H7N9.

Lastly, we tested a suboptimal dose of H7.HK2 combining with the HA2-directed non-neutralizing mAb H7.HK4 against mouse lethal challenge. Compared to HA1 (head domain of HA), the HA2 (stalk domain) is genetically more conserved. Hence, HA2-directed mAbs typically display broader recognition of HA subtypes than HA1-directed mAbs. This is indeed the case for H7.HK4, i.e., in addition to H7N9 and H7N7, it also recognized the HAs from H10N8 and H15N8, to which both H7.HK1 and H7.HK2 had no reactivity. When converted to mouse IgG2a enabling Fc effector function in mice, H7.HK4 demonstrated moderate prophylactic protection at 5 mg/kg and augmented mouse protection of H7.HK2, supporting the inclusion of HA2-directed antibodies in a mAb combination regimen against H7N9.

In summary, from a 2013 H7N9 convalescent case occurring in Hong Kong, we isolated two clonally related HA1-directed neutralizing mAbs H7.HK1 and H7.HK2 that demonstrated prophylactic and therapeutic efficacies in a mouse lethal challenge model. Cryo-EM structures revealed a β14-centered site of vulnerability targeted by H7.HK1 and H7.HK2, those being the first reported antibodies to bind to the H7 lateral patch. Recognition of this conserved epitope facilitates near full binding and neutralization capacity of H7.HK1 and H7.HK2 to the later 2016–2017 H7N9 isolates. This unique epitope at the lateral patch of HA1 renders H7.HK1 and H7.HK2 favorable candidates for combination prophylaxis and therapy against H7N9, which may include multiple HA1-directed neutralizing mAbs targeting different epitopes and benefit from the inclusion of HA2-directed mAbs as well.

## Methods

### Collection of human specimens

A blood specimen was collected from the H7N9_HK2013 patient about one year after recovery from a hospitalized severe H7N9 infection. Written informed consent was obtained from the patient. Each specimen is unique and cannot be replaced once processed. The study was approved by the Institutional Review Board (IRB) of the University of Hong Kong and the Hospital Authority (Reference number: UW-13-265).

### Plasmids, viruses, antibodies, and cells

Expression plasmids encoding the H7 hemagglutinin and N9 neuraminidase based on A/Shanghai/4664T/2013 H7N9 strain were obtained from Dr. Jianqing Xu[39]. Codon-optimized gene encoding the H7 hemagglutinin of A/Hong Kong/125/2017 H7N9 was synthesized (Twist Bioscience) and cloned into pcDNA3.1 (Invitrogen). HIV-1 NL4-3Δenv.Luc.R-E- backbone was obtained through the NIH HIV Reagent Program as contributed by Dr. Nathaniel Landau. These plasmids were used to co-transfect 293T cells to generate H7N9 pseudo viruses. All live replicating influenza A viruses used in this study were isolated from patients and include A/Hong Kong/470129/2013 H7N9[14], A/Zhejiang/DTID-ZJU01/2013 H7N9[3], A/Anhui/1/2013 H7N9 (obtained from the China Center for Disease Control and Prevention), A/Vietnam/1194/2004 H5N1, A/Hong Kong/459094/2010 H5N1, A/Hong Kong/1073/1999 H9N2, A/Hong Kong/415742/2009 H1N1, and A/Hong Kong/400500/2015 H3N2. DNA sequences encoding the variable regions of previous H7N9 mAbs L4A-14 (PDB: 6II4), H7.167 (PDB: 5V2A), and 07-5F01 (GenBank KU987563 and KU987564) were synthesized (Twist Bioscience) and cloned into the corresponding human gamma, kappa, and lambda chain expression vectors described[49–51]. Full IgG1 was expressed by co-transfecting Expi293F cells with equal amounts of paired heavy and light chain plasmids and purified using recombinant protein A agarose (Thermo Fisher). The non-H7N9 placebo mAb used in this study, AD358_n1, has been described[51] and is specific to HIV-1 gp120. Human embryonic kidney 293 cell line, of which the sex is

female, is the parental cell for 293T and Expi293F cell lines. 293T was obtained from ATCC (Cat. No. CRL-11268, Clone 17) and maintained as adherent cells in complete DMEM medium at 37 °C. 293T is highly transfectable and contains SV40 T-antigen. Expi293F was obtained from Thermo Fisher (Cat. No. A14527) and adapted to suspension culture in Expi293 Expression Medium at 37 °C. Madin-Darby Canine Kidney (MDCK) cell line, of which the sex is female, was obtained from ATCC (Cat. No. CCL-34) and maintained as adherent cells in complete DMEM medium at 37 °C. The vendors provided certificates of analysis for the cell lines. No further authentication was performed on cell lines used in this study.

## Single B cell sorting by fluorescence activated cell sorter (FACS)

A soluble recombinant HA ΔTM antigen based on A/Shanghai/2/2013 H7N9 (Immune Technologies, Cat. No. IT-003-0074ΔTMp) was biotinylated via EZ-Link (Thermo Fisher, Cat. No. A39256), followed by streptavidin mediated conjugation of phycoerythrin (PE) (Invitrogen, Cat. No. SA10041). The patient PBMCs were stained with an antibody cocktail to CD3-PE-CF594 (BD Biosciences, Cat. No. 562406, Clone SP34-2), CD19-PE-Cy7 (BioLegend, Cat. No. 302216, Clone HIB19), CD20-APC-Cy7 (BioLegend, Cat. No. 302314, Clone 2H7), IgG-FITC (BD Biosciences, Cat. No. 555786, Clone G18-145), and IgM-V450 (BD Biosciences, Cat. No. 561286, Clone G20-127). In addition, live/dead yellow stain (Invitrogen, Cat. No. L34968) was used to exclude dead cells. All staining antibodies were purchased from vendors providing Quality Certificates and further validated on healthy human blood donor PBMCs purchased from New York Blood Center. After washing, H7-PE$^+$ B cells were sorted using a multi-laser MoFlo sorter (Beckman Coulter, Jersey City, NJ). Fluorescence compensation was performed with anti-mouse Ig kappa CompBeads (BD Biosciences, Cat. No. 552843). Individual B cells were sorted into a 96-well PCR plate, each well containing 20 μL lysis buffer composed of 0.5 μL RNaseOut (Invitrogen, Cat. No. 10777019), 5 μL 5x first-strand buffer, 1.25 μL 0.1 M DTT, and 0.0625 μL Igepal (Sigma, St. Louis, MO). The PCR plate with sorted cells was frozen on dry-ice and then stored at −80 °C. The total cell sample passing through the sorter was analyzed with FlowJo 10.0 (TreeStar, Cupertino, CA).

## Single B cell RT-PCR, sequencing, and cloning

From each sorted cell, the variable regions of IgG heavy and light chains were amplified by RT-PCR and cloned into expression vectors as previously described[51]. Briefly, frozen plates with single B cell RNA were thawed at room temperature, and RT was carried out by adding into each well 3 μL random hexamers at 150 ng/μL (Gene Link, Cat. No. 26-4000-03), 2 μL dNTP (each at 10 mM), and 1 μL SuperScript II (Invitrogen, Cat. No. 18064022), followed by incubation at 42 °C for 2 h. We note that these RT parameters may be suboptimal to those described previously[49,50]. After RT, 25 μL water was added to each well to dilute cDNA, and the cDNA plates were stored at −20 °C for later use. The variable regions of heavy, kappa, and lambda chains were amplified independently by nested PCR in 50 μL, using 5 μL cDNA as template, with HotStarTaq Plus DNA polymerase (Qiagen, Cat. No. 203605) and primer mixes as described[49,52]. Cycler parameters were 94 °C for 5 m, 50 cycles of 94 °C for 30 s, 52–55 °C for 30 s, and 72 °C for 1 m, followed by 72 °C for 10 m. The PCR amplicons were subjected to direct Sanger sequencing, and the antibody sequences were analyzed using IMGT/V-QUEST. Selected PCR sequences that gave productive gamma, kappa, and lambda chain rearrangements were re-amplified with custom primers containing unique restriction digest sites and cloned into the corresponding human gamma, kappa, and lambda chain expression vectors as described[49–51]. Full IgG1 was expressed by co-transfecting Expi293F cells with equal amounts of paired heavy and light chain plasmids and purified using recombinant protein A agarose (Thermo Fisher).

## ELISA and competition ELISA

H7N9 ΔTM HA and HA1 based on A/Shanghai/2/2013, HA1s based on A/Guangdong/17SF003/ 2016, A/Hong Kong/125/2017, and H7N7 ΔTM HA based on A/Netherlands/219/2003 were purchased from Immune Technologies (Cat. No. IT-003-0074ΔTMp, IT-003-0074p, IT-003-0075p, IT-003-0076p, IT-003-0081ΔTMp). Other non-H7 ΔTM HA proteins were purchased from Sino Biological (Cat. No. 40787-V08H, 11688-V08H, 40868-V08B, 40359-V08B, 11720-V08H). ELISA plates were coated with HA or HA1 antigens at 2 μg/mL in PBS at 4 °C overnight. For Endo H treatment, the required amount of antigen was diluted in 10x buffer and mixed with 1 μL Endo H (New England Bio-Labs, Cat. No. P0702S) at 37 °C for 1 h; an equal amount of antigen (untreated) was processed under identical condition without Endo H. Both treated and untreated antigens were diluted in PBS to coat ELISA plates. Coated plates were blocked with 1% BSA (bovine serum albumin) in PBS, followed by incubation with serially diluted mAbs at 37 °C for 1 h. Horseradish peroxidase (HRP)-conjugated goat anti-human IgG Fc (Jackson ImmunoResearch, Cat. No. 109-035-098) was added at 1:10,000 at 37 °C for 1 h. All ELISA incubation volumes were 100 μL/well except that 200 μL/well was used for blocking. Plates were washed between steps with 0.1% Tween 20 in PBS and developed with 3,3′,5,5′-tetramethylbenzidine (TMB) (Novex, Life Technologies), with 1 M $H_2SO_4$ as terminator and read at 450 nm. For competitive ELISA, plates were coated with 2 μg/mL of H7N9 ΔTM HA. After blocking, serial dilutions of competing mAbs were added in 50 μL of blocking buffer, followed by addition of 50 μL of biotin labeled H7.HK2 at 100 ng/mL. After incubation at 37 °C for 1 h, the plates were washed and then incubated with 250 ng/mL of streptavidin-HRP (Jackson ImmunoResearch, Cat. No. 016-030-084) at ambient temperature for 30 m before development with TMB as described above. Similar results were independently reproduced at least once.

## SDS-PAGE and Western blot

Cleaved HA protein from A/Shanghai/1/2013 H7N9 (HA1 + HA2, cleavage) (Sino Biological, Cat. No. 40104-V08H4) was added at 1 μg with 4x SDS loading buffer with reduced reagent and heated at 70 °C for 10 m. The protein was separated on NuPAGE 4-12% Bis-Tris gel with MOPS running buffer (Invitrogen) and transferred to PVDF membrane semi-dry with the Bio-Rad trans-blot turbo transfer system. The membrane was blocked in 2% skim milk in PBS-T, followed by incubation with mAb H7.HK2 or H7.HK4 as primary antibody at 1 μg/mL in blocking buffer at 4 °C overnight. HRP-conjugated goat anti-human IgG Fc (Jackson ImmunoResearch, Cat. No. 109-035-098) was used as secondary antibody at 1:10,000 in blocking buffer at room temperature for 1 h. The immunoreactive band was detected with ECL reagent (Thermo Fisher, Cat. No. A38555). Similar results were independently reproduced at least once.

## H7N9 neutralization assays

H7N9 neutralization was first measured with a single-round infection of MDCK cells using pseudo viruses expressing the H7 gene from A/Shanghai/4664T/2013 H7N9 or A/Hong Kong/125/2017 H7N9, and the N9 gene from A/Shanghai/4664T/2013 H7N9, pseudotyped with the HIV-1 NL4-3Δenv.Luc.R-E- backbone[39]. In 96-well plate, 70 μL of antibody-virus mixture were incubated at 37 °C for 1 h in triplicate wells before transferring to pre-seeded MDCK cells, followed by the addition of 35 μL of medium containing DEAE-dextran at a final concentration of 10 μg/mL. To keep assay conditions constant, sham medium was used in place of antibody in control wells. Infection levels were determined 2 days later with Bright-Glo luciferase assay system (Promega, Cat. No. E1501). Neutralization curves were fitted by a 5-parameter nonlinear regression built in Prism 9.5.1 (GraphPad Software, La Jolla, CA). The 50% inhibitory titers (IC$_{50}$s) were reported as the antibody concentrations required to inhibit infection by 50%. H7N9 neutralization was next measured using live replicating influenza

viruses to infect MDCK cells as described[53]. Briefly, serially diluted mAbs were incubated with 100 $TCID_{50}$ (50% tissue culture infective dose) of an influenza virus at 37 °C for 2 h, and 100 μL virus-mAb mixture was added to MDCK cells. After 1 h incubation, the virus-mAb mixture was removed, and minimum-essential medium with 2 μg/mL L-1-tosylamide-2-phenylethylchloromethyl ketone-treated trypsin (TPCK-trypsin) was added to each well. The plates were then incubated for 72 h, and cytopathic effects were recorded. The mAb concentration that protected 50% of 5 replicate wells from cytopathology was reported as $IC_{50}$. Similar results were independently reproduced at least once.

## H7 HA production
Soluble, disulfide-stabilized, fully cleaved H7 HA trimers were produced by transient co-transfection of plasmids encoding H7 HA (H7 SH13 DS2 6R) and Furin of Expi293F cells (Life Technologies) using Turbo293 transfection Reagent (Speed biosystem). After 5 days at 37 °C, culture supernatants were harvested by centrifugation and concentrated 5-fold by Tangential Flow Filtration. The recombinant HA trimer was captured by Ni-NTA (Sigma-Aldrich) through a C-terminal 6xHis-tag. The imidazole eluant was combined 1:1 (v/v) with saturated ammonium sulfate, centrifuged at 4 °C, and pellet removed. The supernatant was dialyzed against 500 mM NaCl, 50 mM Tris pH 8, and purified by size exclusion chromatography on a Superdex 200 Increase 10/300 GL column (Cytiva).

## Human mAb Fab preparation
Human mAb Fab fragments were produced by digestion of the full IgG antibodies with immobilized Papain (ThermoFisher) equilibrated with 25 mM phosphate, 150 mM NaCl, pH 10, and 2 mM EDTA for 3 h. The resulting Fabs were purified from the cleaved Fc domain by affinity chromatography using protein A. Fab purity was analyzed by SDS-PAGE. All Fabs were buffer-exchanged into 25 mM phosphate, 150 mM NaCl, pH 7.0 prior to cryo-EM experiments.

## Cryo-EM sample preparation, data collection, and structure determination
To determine the structures of H7.HK1 and H7.HK2 with H7 HA trimer, trimer was mixed with the antibody Fab at 1 to 1.2 molar ratio at a final total protein concentration of ~1 mg/mL and adjusted to a final concentration of 0.005% (w/v) n-Dodecyl β-D-maltoside (DDM) to prevent preferred orientation and aggregation during vitrification. Cryo-EM grids were prepared by applying 3 μL of sample to a freshly glow discharged carbon-coated copper grid (CF 1.2/1.3 300 mesh). The sample was vitrified in liquid ethane using a Vitrobot Mark IV with a wait time of 30 s, a blot time of 3 s, and a blot force of 0. Cryo-EM data were collected on a Titan Krios operating at 300 keV, equipped with a K3 detector (Gatan) operating in counting mode. Data were acquired using Leginon[54]. The dose was fractionated over 50 raw frames. For all structures, the movie frames were aligned and dose-weighted[55] using cryoSPARC 3.4[56]; the CTF estimation, particle picking, 2D classifications, ab initio model generation, heterogeneous refinements, homogeneous 3D refinements and non-uniform refinement calculations were carried out using cryoSPARC 3.4[56].

## Atomic model building and refinement
For structural determination, a model of the antibody Fab was generated using SAbPred[57]. The Fab model and the crystal structure of an H7 HA mutant (PDB: 6IDD)[10] was docked into the cryo-EM density map using UCSF Chimera[58] to build an initial model of the complex. The model was then manually rebuilt to the best fit into the density using Coot[59] and refined using Phenix[60]. Interface calculations were performed using PISA[61]. Structures were analyzed and figures were generated using PyMOL (http://www.pymol.org) and UCSF Chimera[58]. Final model statistics are summarized in Table S1.

## H7 HA sequence analysis
Searching the Global Initiative on Sharing All Influenza Data (GISAID) with H7Nx returned a total of $N = 1511$ H7 HA sequences, with $N = 2$ H7N2, $N = 2$ H7N3, $N = 2$ H7N4, $N = 54$ H7N7, and $N = 1451$ H7N9. After removing $N = 28$ duplicate or defective sequences, $N = 1483$ H7 amino acid sequences were aligned using Clustal[62], and the sequence conservation analysis was performed using AL2CO[63].

## Mouse prophylactic and therapeutic studies
The mouse prophylactic and therapeutic studies were approved by the Committee on the Use of Live Animals in Teaching and Research (CULATR) of the University of Hong Kong (Reference number: 4011-16) and conducted in biosafety level 3 animal facilities as described previously[64–66]. Female BALB/c mice were imported from Harlan UK Ltd, UK, and those of 6–8 weeks of age were obtained from the Laboratory Animal Unit of the University of Hong Kong. Mice were housed at temperatures between 22 to 25 °C with dark/light cycles and given access to standard pellet feed and water ad libitum. For prophylactic study, one day before virus inoculation, each mouse was administered with 100 μL of mAb at 1 mg/mL intraperitoneally. For therapeutic study, infected mice were administered with 100 μL of mAb at 1 mg/mL intraperitoneally at day 1 post viral challenge. Mice in the control groups were administered with either PBS or with a non-H7N9 mAb. On the day of virus infection, each mouse was inoculated with 10 $LD_{50}$ (40 μL) of H7N9/AH1 virus through intranasal route. Virus inoculation was performed under ketamine (100 mg/kg) and xylazine (10 mg/kg) anesthesia. The mice were monitored for 14 days with disease severity score and body weight recorded daily. Disease severity were scored as follow: Score 0, apparently healthy; Score 1, mild disease symptom with ruffled fur but still active; Score 2, medium disease symptom with ruffled fur, reduced activity and no weight gain; Score 3, severe disease symptoms with ruffled fur, hunched posture, labored breathing and weight loss; Score 4, moribund being very inactive, showing difficulty moving around and accessing to food and water, and weight loss. The predefined humane endpoints were either a weight loss of ≥20% or a disease severity score of 4. Mice were euthanized if the humane endpoints were reached. Each study group included 5–15 randomly allocated female mice for calculating survival rates and statistically significant differences based on the viral challenge model previously established with only female mice[64–66]. The investigators were blinded to animal group allocation during data collection.

## Statistical analysis
GraphPad Prism 9.5.1 was used to plot the ELISA data using sigmoidal dose-response with variable slope for curve fitting and neutralization data using 5-parameter nonlinear regression for curve fitting. All quantitative data are presented as mean ± standard error (SEM). GraphPad Prism 9.5.1 was used to plot the mouse survival curves. Two-sided unpaired student's t-test in GraphPad Prism 9.5.1 and Microsoft 365 Excel version 2404 was used for comparisons between mouse groups. $P$ values < 0.05 were considered statistically significant; $P$ values of 0.05-0.10 were considered statistical trends.

## Reporting summary
Further information on research design is available in the Nature Portfolio Reporting Summary linked to this article.

# Data availability
Sequences of the heavy and light chain variable regions of the four H7N9 human mAbs have been deposited in GenBank under accession # OR901962 – OR901969. The cryo-EM reconstruction of H7.HK1 and H7.HK2 Fabs in complex with H7 SH13 DS2 6R HA has been deposited in the Electron Microscopy Data Bank as EMD-41422 and EMD-41441 and the Protein Data Bank as PDB: 8TNL and 8TOA. Materials will be

made available to researchers with appropriate materials transfer agreements (MTAs). All inquiries should be sent to the corresponding authors. Source data are provided with this paper.

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

## Acknowledgements

We thank the patient for donating blood for the study. We thank Reda Rawi and Jeffrey C. Boyington for design of H7 SH13 DS2 6R used for structural analysis. Cryo-EM data were collected at the Columbia University Cryo-Electron Microscopy Center. We thank Shuofeng Yuan and Vincent Poon for assistance with the animal experiments. This study is supported by U.S. Department of Defense contract No. W911NF-14-C-0001 to D.D.H. and X.W., by Health@InnoHK, Innovation and Technology Commission of Hong Kong to K.Y. and K.K.T., by donations from Richard Yu and Carol Yu, Shaw Foundation Hong Kong, Michael Seak-Kan Tong, The Hui Ming, Hui Hoy and Chow Sin Lan Charity Fund Limited, Chan Yin Chuen Memorial Charitable Foundation, Marina Man-Wai Lee, Jessie and George Ho Charitable Foundation, Kai Chong Tong, Tse Kam Ming Laurence, Foo Oi Foundation Limited, Betty Hing-Chu Lee, and Ping Cham So to K.Y. and K.K.T., by Bill and Melinda Gates Foundation grants FNIH SHAP19IUFV and INV-016167 to L.S., and by U.S. National Institutes of Health, National Institute of Allergy and Infectious Disease, Intramural Research Program of the Vaccine Research Center ZIA IA0005022 to P.D.K.

## Author contributions

Conceptualization: X.W., D.D.H., and K.Y. Methodology: X.W., K.K.T., and L.S. Investigation: M.J., H.Z., N.C.M., H.L., Y.L., H.D., and J.E.B. Visualization: X.W. and N.C.M. Funding acquisition: D.D.H., X.W., K.Y., K.K.T., L.S., and P.D.K. Project administration: X.W. and K.K.T. Supervision: X.W., K.K.T., K.Y., P.D.K., and L.S. Writing—original draft: X.W., K.K.T., and N.C.M. Writing—review & editing: X.W., K.K.T., M.J., H.Z., N.C.M., K.Y., D.D.H., P.D.K., and L.S.

## Competing interests

An U.S. provisional patent titled "Human Protective Neutralizing and Non-neutralizing Antibodies and Their Use against Influenza Viruses" was filed with filing No. 63/650,342 and X.W., M.J., N.C.M., H.L., D.D.H., K.Y., K.K.T., and L.S. as co-inventors; the remaining authors declare no competing interests. The authors declare no other competing interests.
