## [Peer Review File · Nature Communications]

Human Neutralizing Antibodies Target a Conserved Lateral Patch on H7N9 Hemagglutinin HeadReviewers' Comments:

Reviewer #1:

Remarks to the Author:

The mechanisms by which antibodies neutralize and protect against viral infection remains an important focus of vaccinologist. In immunologically and structurally characterizing monoclonal antibodies that target H7 hemagglutinin, Jia and colleagues revealed how two antibodies work in concert to increase their antiviral potency in vivo (mouse model). It reflects how an optimal polyclonal antibody response may work to protect against disease by targeting different regions of the same molecule. It also highlights advantages of using a cocktail of monoclonal antibodies as a therapeutic.

The strength of this study shows the robustness of protection of mAbs H7K1 and H7.HK2 in pre-exposure prophylactic experiments against live H7N9 challenges and, through structural studies, where the mAbs recognize the beta14-centered surface of H7 HA1. Moreover, antigenic changes in the globular head in more recent years (2016 to 2017) have rendered a lot of published neutralizing mAbs against ineffective in recognizing (although, this should have been demonstrated in the present study). The finding that both H7.HK1, H7.HK2 and the HA2 mAb H7.HK4 are still effective against more contemporary strains is promising.

However, there are several weaknesses in the study that should be noted. The current work does not really add any significant insight into novel mechanisms of antibody inhibitory activity. There are already data to suggest and show that combination therapy can result in more pronounced protection in vivo. Post-exposure prophylactic experiments (one day post infection) are not robust enough to warrant its use as a therapeutic for either H7.HK1, H7.HK2 or H7HK.4. Given the inability of generating escape mutants against non-neutralizing antibodies that protect via Fc-mediated immunity, it would have been great if the present study had structurally resolved how H7HK.4 bound to the HA2 region – this would have increased the impact of this paper. It would also have benefited the reader if the authors had included past published (if they had access to the sequence or reached out to other authors) mAbs and show that they do not recognize more contemporary strains of H7 in the current study. Lastly, there is weak data to suggest there is 'allosteric' neutralization that is occurring. It is not clear if 'allosteric' is the best term to use in this case. One could argue that the only data is an in vivo experiment where two mAbs were mixed together to provide protection – no neutralization or structural experiment to warrant using the term allosteric.

Other comments:

Page 2; line 17: The data as presented arguably does not show allosteric mechanism of mAb neutralization. However, there is data to support augmented protection when mAbs are given as a cocktail.

Page 4; line 10: The phrase 'HA2-directed mAbs typically lacked neutralizing activity' is arguably not accurate. Data generated from multiple investigators/groups in the past decade have clearly demonstrated that a large number of (murine and human) HA2 mAbs do have neutralizing activity against divergent subtypes. What is probably more typical is that HA2 mAbs are not as robust in neutralizing activity when compared to HA1 mAbs. The sentence should be rephrased.

Page 10; line 19: The authors might mean, 'this analysis is consistent with'

Reviewer #2:

Remarks to the Author:

The authors discovered and characterized four human monoclonal antibodies, named H7.HK1, H7.HK2, H7.HK3 and H7/HK4, from a convalescent patient infected with A/Hong Kong/470129/2013 (H7N9) virus that the authors reported before (ref. 14). IgG B-cells of the PBMC samples from the patient were sorted against a soluble recombinant H7 HA protein from A/Shanghai/2/2013 (H7N9),

and finally these four antibodies were recovered for further studies.

Measured by ELISA, all four antibodies were found to bind to full-length H7 HA ectodomain. All antibodies except H7.HK4 also bound H7 HA head, and it was proposed H7.HK4 likely bound to stem region of H7 HA. H7.HK3 cross-reacted with an H15 HA, and H7.HK4 cross-reacted with H10 and H15 HAs.

Neutralization studies were conducted with H7N9 pseudo-viruses and live replicating H7N9 viruses. H7.HK1 and H7.HK2 which shared the same VH and VL germline genes as well as the same CDR lengths neutralized H7N9 viruses isolated in original 2013 and later 2016 strains. But H7.HK3 and H7.HK4 did not show any H7N9 neutralization under experimental conditions. All four antibodies did not neutralize the tested H3N2, H1N1, H5N1 and H9N2 viruses.

Using cryo-EM, Fabs of H7.HK1 and H7.HK2 with H7 HA ectodomain from A/Shanghai/2/2013 (H7N9) were determined to 3.62 and 3.69 Å resolution, respectively. Both antibodies bound to similar H7 epitopes that are located in the HA head but away from HA receptor binding site. It was interesting that upon H7.HK1 and H7.HK2 binding to the H7 HA, the 220-loop from an adjacent H7 HA protomer became disordered under cryo-EM experiment conditions. Based on this finding, the authors proposed an allosteric mechanism of neutralization employed by these two antibodies.

In mouse models for prophylactic and therapeutic studies, H7.HK1 and H7.HK2 protected H7N9 virus infection, while H7.HK3 and H7.HK4 showed no protection. Interestingly, for H7.HK4 which is human IgG1, the engineered antibody H7.Hk4.mIgG2a with a mouse Fc region showed protection against H7N9 virus.

Overall, this is a systematic study of human antibodies against H7N9 viruses, and antibodies H7.HK1 and H7.HK2 were proposed to be one of the best human antibodies for neutralization potency and mouse protection efficacy, as well as their breath against the original 2013 H7N9 virus and recent 2016 H7N9 viruses.

The major points:

1. Title "Allosteric Neutralization by Human H7N9 Antibodies". In cryo-EM structures, upon H7.HK1 and H7.HK2 binding to H7HA protomer, the 220-loop of an adjacent protomer became disordered under cryo-EM conditions and the antibodies might clash with the 220-loop if it was ordered. But more evidence appears to be needed to claim allosteric neutralization by these two antibodies. First of all, does the H7 HA bound with H7.HK1 or H7.HK2 still bind sialic acid receptors? Since the disordered 220-loop is in an adjacent protomer and with the documented information about HA heads being able to open and close in a breathing motion, such when binding FluA-20 binds at the HA head trimer interface, the 220-loop might still be functional for receptor binding in a temporal manner. It may also be possible that steric block of HA binding to receptors on the cell surface could contribute to virus neutralization.

2. Abstract, Page 2, line 17-19: "Our data demonstrated an allosteric mechanism of mAb neutralization and augmented protection against H7N9 when a HA1-directed neutralizing mAb and a HA2-directed non-neutralizing mAb were combined". In addition to the question of whether an allosteric mechanism is proved, the augmented protection may need to be caveated against H7N9 by H7.HK2 plus H7.HK4 (Fig. 3). Actually, in mouse prophylactic experiments, the mouse survival rate and body weight loss were about the same for H7.HK2 plus H7.HK4.mIgG2a vs H7.HK2 itself (Fig. 3A), although the combination of H7.HK2 and H7.HK4.mIgG2a was a little better than the control combination of H7.HK2 and H7.HK4.mIgG1 with different mouse Fc regions.

3. The binding epitopes of H7.HK1 and H7.HK2 are actually on H7 HA lateral patch, as recently proposed in H1 HA structure (PMID 29255041, Raymond D.D. et al, PNAS, 2018, 115(1): 168-173).

The authors may focus the discussion on this novel H7 lateral patch epitope, and potency and breadth of H7.HK1 and H7.HK2.

Minor points:

1. Neuraminidase inhibitors Tamiflu and Relenza as well as endonuclease inhibitor Xofluza were approved to treat influenza infection, and were tested to be effective for H7N9 viruses. Since human H7N9 infection cases are rare, these might not yet have been reported for treatment efficacy. While antibody research for H7N9 virus is very important, these small molecule inhibitors are based on different targets. Accordingly, the relevant introduction in this manuscript could be revised.

2. For clarity and comparison, please use H3 numbering for the H7 HA and Kabat numbering for the antibodies.

3. Page 7, Table 1. Please add to the notes full strain names of all viruses.

4. Fig. 2. Figs. 2D and 2E could be made clearer by including only key residues and excluding unrelated secondary structures and others.

5. Page 23, line 17. "GenBank under accession # xxxxxxxx to xxxxxxxx" Please add.

6. Page 24, Reference 6. Incomplete journal reference. Please add page number.

7. Page 28. Please be consistent in use of abbreviations for the journal names.

8. Supplementary Fig. 2. "(F) Representative density is shown for the interface of H7.HK1 heavy chain, light chain, and H7 HA. (G) Representative density is shown for the interface of H7.HK2 heavy chain, light chain, and H7 HA." Please specify what chains are in the different colors.

9. Page 38. Supplementary Table 1. "Electron exposure ($e^{-}/\text{\AA}^2$)" Please replace with " $e^{-}/\text{\AA}^2$ "

10. Page 38. Supplementary Table 1. Please truncate B-values to integers as decimal points not meaningful. Is the ligand cited here the antibody Fab? If so, please indicate as antibody or Fab. For the second entry, the B value of the Fab is lower than the HA? This would be unusual as the constant regions of the Fab are usually disordered as indicated in fig. S2E.

Point-by-Point Response to the Reviewers

Reviewer #1:

The mechanisms by which antibodies neutralize and protect against viral infection remains an important focus of vaccinologist. In immunologically and structurally characterizing monoclonal antibodies that target H7 hemagglutinin, Jia and colleagues revealed how two antibodies work in concert to increase their antiviral potency in vivo (mouse model). It reflects how an optimal polyclonal antibody response may work to protect against disease by targeting different regions of the same molecule. It also highlights advantages of using a cocktail of monoclonal antibodies as a therapeutic.

Response: We appreciate the Reviewer's overall positive view of our study.

The strength of this study shows the robustness of protection of mAbs H7.HK1 and H7.HK2 in pre-exposure prophylactic experiments against live H7N9 challenges and, through structural studies, where the mAbs recognize the beta14-centered surface of H7 HA1. Moreover, antigenic changes in the globular head in more recent years (2016 to 2017) have rendered a lot of published neutralizing mAbs against ineffective in recognizing (although, this should have been demonstrated in the present study). The finding that both H7.HK1, H7.HK2 and the HA2 mAb H7.HK4 are still effective against more contemporary strains is promising.

Response: We agree with the Reviewer and added 3 previous mAbs, L4A-14, H7.167, and 07-5F01, for direct comparison with H7.HK2 (Results, page 7-8, Supplement Fig. 1). We corrected some mistakes and more accurately determined the mAb neutralization IC₅₀s (Fig. 1d and Table 1). Since the GD2016 pseudo virus did not generate high enough titers, we used the HK2017 H7N9 pseudo virus for neutralization comparison. Evaluated both by H7 antigen binding and pseudo virus neutralization, H7.HK2 is superior to the two best RBS-directed mAbs L4A-14 and H7.167 and matches the one best non-RBS mAb 07-5F01 against H7N9.

However, there are several weaknesses in the study that should be noted. The current work does not really add any significant insight into novel mechanisms of antibody inhibitory activity. There are already data to suggest and show that combination therapy can result in more pronounced protection in vivo. Post-exposure prophylactic experiments (one day post infection) are not robust enough to warrant its use as a therapeutic for either H7.HK1, H7.HK2 or H7HK.4. Given the inability of generating escape mutants against non-neutralizing antibodies that protect via Fc-mediated immunity, it would have been great if the present study had structurally resolved how H7HK.4 bound to the HA2 region – this would have increased the impact of this paper. It would also have benefited the reader if the authors had included past published (if they had access to the sequence or reached out to other authors) mAbs and show that they do not recognize more contemporary strains of H7 in the current study. Lastly, there is weak data to suggest there is 'allosteric' neutralization that is occurring. It is not clear if 'allosteric' is the best term to use in this case. One could argue that the only data is an in vivo experiment where two mAbs were mixed together to provide protection – no neutralization or structural experiment to warrant using the term allosteric.

Response: We appreciate the Reviewer's view on the less studied HA2-directed antibodies. We confirmed the H7.HK4 epitope on HA2 by Western blot (Fig. 1c). We have also attempted to obtain the cryoEM structures of H7.HK3 and H7.HK4, but those Fabs did not bind the stabilized H7 trimer under typical cryoEM conditions. We have added the ELISA binding data of all four mAbs to the stabilized H7 trimer in Supplementary Fig. 3a. H7.HK4 did not bind the stabilized H7 trimer, likely explaining its lack of neutralization. For the two HA1-directed neutralizing mAbs H7.HK1 and H7.HK2, their epitopes are distinct from previous H7N9 neutralizing mAbs. Hence, these mAbs are valuable to complement other neutralizing antibodies for combination therapy. Regarding previous mAbs, we have included three for direct comparison with H7.HK2 – see response above. We also agree with the Reviewer that our data does not warrant using the term “allosteric” and have thus removed it.

Other comments:

Page 2; line 17: The data as presented arguably does not show allosteric mechanism of mAb neutralization. However, there is data to support augmented protection when mAbs are given as a cocktail. – We agree and removed “allosteric” from Abstract.

Page 4; line 10: The phrase ‘HA2-directed mAbs typically lacked neutralizing activity’ is arguably not accurate. Data generated from multiple investigators/groups in the past decade have clearly demonstrated that a large number of (murine and human) HA2 mAbs do have neutralizing activity against divergent subtypes. What is probably more typical is that HA2 mAbs are not as robust in neutralizing activity when compared to HA1 mAbs. The sentence should be rephrased. – We agree and rephrased the sentence to: HA2-directed mAbs have also demonstrated neutralizing activity against divergent subtypes, although typically not as robust in neutralizing activity when compared to HA1 mAbs (page 4).

Page 10; line 19: The authors might mean, ‘this analysis is consistent with’ – Yes, we revised the sentence to “this analysis is consistent with” (page 12).

Reviewer #2:

The authors discovered and characterized four human monoclonal antibodies, named H7.HK1, H7.HK2, H7.HK3 and H7.HK4, from a convalescent patient infected with A/Hong Kong/470129/2013 (H7N9) virus that the authors reported before (ref. 14). IgG B-cells of the PBMC samples from the patient were sorted against a soluble recombinant H7 HA protein from A/Shanghai/2/2013 (H7N9), and finally these four antibodies were recovered for further studies.

Measured by ELISA, all four antibodies were found to bind to full-length H7 HA ectodomain. All antibodies except H7.HK4 also bound H7 HA head, and it was proposed H7.HK4 likely bound to stem region of H7 HA. H7.HK3 cross-reacted with an H15 HA, and H7.HK4 cross-reacted with H10 and H15 HAs.

Neutralization studies were conducted with H7N9 pseudo-viruses and live replicating H7N9 viruses. H7.HK1 and H7.HK2 which shared the same VH and VL germline genes as well as the

same CDR lengths neutralized H7N9 viruses isolated in original 2013 and later 2016 strains. But H7.HK3 and H7.HK4 did not show any H7N9 neutralization under experimental conditions. All four antibodies did not neutralize the tested H3N2, H1N1, H5N1 and H9N2 viruses.

Using cryo-EM, Fabs of H7.HK1 and H7.HK2 with H7 HA ectodomain from A/Shanghai/2/2013 (H7N9) were determined to 3.62 and 3.69 Å resolution, respectively. Both antibodies bound to similar H7 epitopes that are located in the HA head but away from HA receptor binding site. It was interesting that upon H7.HK1 and H7.HK2 binding to the H7 HA, the 220-loop from an adjacent H7 HA protomer became disordered under cryo-EM experiment conditions. Based on this finding, the authors proposed an allosteric mechanism of neutralization employed by these two antibodies.

In mouse models for prophylactic and therapeutic studies, H7.HK1 and H7.HK2 protected H7N9 virus infection, while H7.HK3 and H7.HK4 showed no protection. Interestingly, for H7.HK4 which is human IgG1, the engineered antibody H7.Hk4.mIgG2a with a mouse Fc region showed protection against H7N9 virus.

Overall, this is a systematic study of human antibodies against H7N9 viruses, and antibodies H7.HK1 and H7.HK2 were proposed to be one of the best human antibodies for neutralization potency and mouse protection efficacy, as well as their breadth against the original 2013 H7N9 virus and recent 2016 H7N9 viruses.

Response: We appreciate the Reviewer's detailed review and summary of our study.

The major points:

1. Title "Allosteric Neutralization by Human H7N9 Antibodies". In cryo-EM structures, upon H7.HK1 and H7.HK2 binding to H7HA protomer, the 220-loop of an adjacent protomer became disordered under cryo-EM conditions and the antibodies might clash with the 220-loop if it was ordered. But more evidence appears to be needed to claim allosteric neutralization by these two antibodies. First of all, does the H7 HA bound with H7.HK1 or H7.HK2 still bind sialic acid receptors? Since the disordered 220-loop is in an adjacent protomer and with the documented information about HA heads being able to open and close in a breathing motion, such when binding FluA-20 binds at the HA head trimer interface, the 220-loop might still be functional for receptor binding in a temporal manner. It may also be possible that steric block of HA binding to receptors on the cell surface could contribute to virus neutralization.

Response: We agree with the Reviewer's interpretation of our data and appreciate the likely mechanism of "steric occlusion". We have thus removed the term "allosteric".

2. Abstract, Page 2, line 17-19: "Our data demonstrated an allosteric mechanism of mAb neutralization and augmented protection against H7N9 when a HA1-directed neutralizing mAb and a HA2-directed non-neutralizing mAb were combined". In addition to the question of whether an allosteric mechanism is proved, the augmented protection may need to be caveated against H7N9 by H7.HK2 plus H7.HK4 (Fig. 3). Actually, in mouse prophylactic experiments, the mouse survival rate and body weight loss were about the same for H7.HK2 plus

H7.HK4.mIgG2a vs H7.HK2 itself (Fig. 3A), although the combination of H7.HK2 and H7.HK4.mIgG2a was a little better than the control combination of H7.HK2 and H7.HK4.mIgG1 with different mouse Fc regions.

Response: We appreciate the Reviewer's careful assessment of our data. In Fig. 3a, we have now overlaid the survival and body weight data for H7.HK2 by itself and in combination (Page 14). The mAb combination did not improve the body weight trough as both groups fully protected mice from death with up to 7-8% weight loss. The mAb combination did improve the recovery of weight loss starting on day 7 post challenge (Fig. 3a, 4th row). We acknowledge the degree of improvement was moderate and noted it so both in the Abstract (page 2) and Discussion (page 19).

3. The binding epitopes of H7.HK1 and H7.HK2 are actually on H7 HA lateral patch, as recently proposed in H1 HA structure (PMID 29255041, Raymond D.D. et al, PNAS, 2018, 115(1): 168-173). The authors may focus the discussion on this novel H7 lateral patch epitope, and potency and breadth of H7.HK1 and H7.HK2.

Response: We truly appreciate the Reviewer's suggestion to focus on the epitopes of H7.HK1 and H7.HK2 that target the lateral patch of HA head initially identified with H1. We have now added Fig. 2g for the lateral patch analysis and a new Fig. 4 to define the HA1 lateral patch as a supersite for neutralizing antibodies (Results, page 14-15) and discussed this site of vulnerability (page 17-18).

Minor points:

1. Neuraminidase inhibitors Tamiflu and Relenza as well as endonuclease inhibitor Xofluza were approved to treat influenza infection, and were tested to be effective for H7N9 viruses. Since human H7N9 infection cases are rare, these might not yet have been reported for treatment efficacy. While antibody research for H7N9 virus is very important, these small molecule inhibitors are based on different targets. Accordingly, the relevant introduction in this manuscript could be revised. – We appreciate this input and revised the relevant Introduction to include the small molecule inhibitors and removed “lack of treatment efficacy”.

2. For clarity and comparison, please use H3 numbering for the H7 HA and Kabat numbering for the antibodies. – We have revised the manuscript to use H3 numbering for H7 HA and Kabat numbering for antibodies.

3. Page 7, Table 1. Please add to the notes full strain names of all viruses. – Table 1 has been modified to include full names of all viral strains.

4. Fig. 2. Figs. 2D and 2E could be made clearer by including only key residues and excluding unrelated secondary structures and others. – Figs. 2d and 2e have been modified to include only key residues.

5. Page 23, line 17. "GenBank under accession #xxxxxxx to xxxxxxx" Please add. – GenBank accession numbers have been added.

6. Page 24, Reference 6. Incomplete journal reference. Please add page number. – Reference 6 has been updated to include the page number.

7. Page 28. Please be consistent in use of abbreviations for the journal names. – References have been updated to be consistent in journal name abbreviations.

8. Supplementary Fig. 2. "(F) Representative density is shown for the interface of H7.HK1 heavy chain, light chain, and H7 HA. (G) Representative density is shown for the interface of H7.HK2 heavy chain, light chain, and H7 HA." Please specify what chains are in the different colors. – We have now specified the antibody chains.

9. Page 38. Supplementary Table 1. "Electron exposure ($e^{-}/\text{Å}^2$)" Please replace with " $e^{-}/\text{Å}^2$ " – Done.

10. Page 38. Supplementary Table 1. Please truncate B-values to integers as decimal points not meaningful. Is the ligand cited here the antibody Fab? If so, please indicate as antibody or Fab. For the second entry, the B value of the Fab is lower than the HA? This would be unusual as the constant regions of the Fab are usually disordered as indicated in fig. S2E. – B-values are now displayed as integers. "Ligand" here referred to N-linked glycans, not to Fab. This has been corrected.

Reviewers' Comments:

Reviewer #1:

Remarks to the Author:

The authors had done a very good job of addressing past concerns/suggestions to their manuscript. they have also done a very good job in addressing the second reviewer's concerns. Thank you.

Reviewer #2:

Remarks to the Author:

My previous comments have been satisfactorily addressed.